# Nonbacterial Thrombotic Endocarditis Associated with Acute Promyelocytic Leukemia: An Autopsy Case Report

**DOI:** 10.3390/medicina57111264

**Published:** 2021-11-18

**Authors:** Tadayuki Hashimoto, Tatsuya Aoki, Yoshitaka Kawabata, Yoshihiro Owai, Yoshikazu Matsuda, Shinobu Tamura

**Affiliations:** 1Department of General Internal Medicine, Hashimoto Municipal Hospital, Hashimoto 648-0005, Japan; t.aoki0914@gmail.com; 2Hinata Medical Clinic, Osaka 543-0062, Japan; kawabata.1023@gmail.com; 3Department of Neurosurgery, Hashimoto Municipal Hospital, Hashimoto 648-0005, Japan; owaiyo@icloud.com; 4Showa University Fujigaoka Hospital, Yokohama 227-8501, Japan; ymatsuda0517@yahoo.co.jp; 5Department of Hematology/Oncology, Wakayama Medical University, Wakayama 641-8509, Japan; stamura@wakayama-med.ac.jp

**Keywords:** nonbacterial thrombotic endocarditis (NBTE), leukemic cell, acute promyelocytic leukemia, anticoagulation therapy

## Abstract

Valve vegetation is one of the most fearful findings for physicians. The first diagnosis that comes to their mind is infective endocarditis (IE), but it can also be noninfective; nonbacterial thrombotic endocarditis (NBTE). NBTE can be even more challenging than IE for physicians because of the wide range of differential diagnoses such as malignancies, autoimmune disorders and human immunodeficiency virus. A 45-year-old woman presented at the emergency room with a sudden onset of dysarthria and right-sided hemiplegia. Laboratory data showed her blood counts and coagulation test were mostly normal and the magnetic resonance imaging detected a high-signal-intensity change in her left brain. An echocardiogram found a vegetation-like structure on her atrial valve. We highly suspected IE leading to cerebral embolism. The clot was successfully removed by our neurosurgeons and anticoagulation therapy was started concurrently. Her state of consciousness improved, but then she suffered a brain hemorrhage and died. The autopsy revealed that the cause of her vegetation was acute promyelocytic leukemia (APL). Based on these findings, it is important to remember that APL can be the cause of NBTE even if the blood count and coagulation tests are almost normal.

## 1. Introduction

Nonbacterial thrombotic endocarditis (NBTE) is an uncommon condition characterized by sterile noninfectious vegetation on the valvular surface of the heart [1,2]. In 1888, Zeigler et al. first introduced NTBE as thromboendocarditis with the deposition of fibrin on cardiac valves [1,2]. Subsequently, it was described in 1936 when Gross and Friedberg named the term “non-bacterial thrombotic endocarditis” [1]. Over the years, NTBE remains a serious and potentially life-threatening disease because of embolism to any organs including the brain and the heart, but it is very difficult to diagnose and manage each case as yet. Many cases of NTBE are underdiagnosed and identified by postmortem examinations. Indeed, valve vegetations caused by NBTE have been reported in approximately 1.2% of all autopsy patients [3].

NTBE is frequently associated with underlying diseases such as malignancies, autoimmune disorders and human immunodeficiency virus (HIV) [1,2]. The incidence of NBTE in patients with advanced malignancies was reported as 19% and most frequently with adenocarcinoma [1,2]. A hypercoagulable state in such cases occurs due to the ability of tumor cells to activate the coagulation system. Meanwhile, the reports of NTBE associated with hematological malignancies are limited, frequently despite of hypercoagulable states. We experienced a fatal case of cerebral infarction caused by NBTE with a sudden onset of neurological symptoms. To the best of our knowledge, we present the first case with NTBE in the world where the diagnosis of acute promyelocytic leukemia (APL) was diagnosed by autopsy.

## 2. Case Presentation

A 45-year-old female Asian nurse with a history of well-controlled hyperthyroidism presented at the emergency room with a sudden onset of dysarthria and paralysis at the weekend. She complained of feeling ill a week before the first visit but had been able to go about her daily life and work. Physical examination revealed a fever over 38.0 °C, altered mental status (13 points on Glasgow coma scale), drop of the right mouth, and paralysis of the right upper and lower extremity. She had not received dental or surgical procedures for one year before the visit.

Table 1 presents our patient’s laboratory data at the first visit. While hemoglobin and platelet count were slightly low, white blood cell count was normal (Table 1). The prothrombin time (PT) was slightly prolonged, and the activated partial thromboplastin time (APTT) was within the normal range. Renal and liver function tests were normal. Lactate dehydrogenase and C-reactive protein levels were elevated. The leukocyte fraction and the other coagulation tests could not be carried out in our hospital due to it being the weekend. Unfortunately, the clinical laboratory technicians who usually observe blood cells professionally were not available during the weekend.

Computed tomography (CT) of the head did not show any abnormal findings including an early CT sign. The magnetic resonance imaging (MRI) detected a high-signal-intensity change in her left brain, which indicated M1 occlusion (Figure 1A,B). Electrocardiography showed normal sinus rhythm. An echocardiogram revealed a vegetation-like structure on her atrial valve. We strongly suspected infectious endocarditis (IE) complicated by cerebral embolism. Therefore, a blood culture was taken immediately, and empirical antibiotics were administered. Anticoagulation therapy was commenced concurrently, and our neurosurgeons decided to perform an endovascular thrombectomy. The blood clot was successfully removed (Figure 1C insert), leading to rapid and complete reperfusion (Figure 1C). Soon after, the level of her consciousness improved, and she was admitted into the department of neurosurgery at our hospital.

However, six hours after the endovascular thrombectomy, the level of her consciousness worsened markedly again, and computed tomography revealed massive intracranial hemorrhage around the stroke area. We judged the clinical diagnosis of brain death on the next day, and she died on the fourth hospital day. Regardless of suspecting IE, no bacteria were found in the thrombus obtained from the cerebral artery, and these blood cultures were negative. At the time of her death, leukemia cells were first reported in the peripheral blood (Figure 2) after the weekend. The abnormal leukocytes had abundant azure granules, strong nuclear irregularities, and many Auer bodies (Figure 2, arrow) in the peripheral blood. A pathological autopsy was performed shortly after death with written informed consent from the patient’s family.

## 3. Autopsy

The autopsy revealed vegetation attached to the atrial valve (Figure 3A,B). Microscopic examination suggested that it was an aggregate of abnormal leukocyte cells and a thrombus (Figure 3C). The thrombus aggregated from abnormal cells were similarly found in many arteries of the brain, the lung, the liver, the spleen and the kidney.

The pathological finding of the femoral bone marrow showed markedly hypercellular with predominantly abnormal cells (Figure 3D). Immunohistochemistry demonstrated diffuse myeloperoxidase and MIB-1 positivity (Figure 3E,F), but CD34 and c-Kit negativity. In the bone marrow specimen at autopsy, flow cytometry revealed an abnormal myeloid blast population highly expressing CD13 and CD33 with no expression of HLA-DR, CD4, CD8, CD14, CD19, CD34, CD41 or CD56. Moreover, we could detect the PML/RARA fusion gene characteristic of APL in the bone marrow samples at the postmortem examinations. Therefore, our patient was finally diagnosed with NBTE-induced stroke and multiple organ emboli caused by APL.

## 4. Discussion

Although the prevalence of NBTE in patients with malignancies remains unknown, it has been described that cardiac valvular vegetation occurred in approximately 20% of patients with advanced stage malignancies [4]. Therefore, it is particularly important to suspect NBTE in patients with malignancies developing cerebral infarction. Adenocarcinoma of the lung, the pancreas, or the stomach is histologically frequently associated with NBTE, which is considered as a part of the clinical spectrum of Trousseau syndrome [5]. Infective endocarditis (IE) is the most important differential diagnosis for NBTE. However, the vegetation in NBTE is much smaller than that in IE, often less than 3 mm [4]. Therefore, NBTE is difficult to diagnose on transthoracic echocardiography (TTE) and is often discovered as an unexpected finding at autopsy. Moreover, more patients with NTBE developed multiple cerebral infraction than those with IE [6]. Disseminated intravascular coagulopathy (DIC) was found in more than 70% of cancers developing NBTE [7]. The pathogenesis was expected to be associated with the activation of thrombosis and fibrinolysis induced by DIC.

Large clinical studies related to NTBE are limited because of its rarity. In 2020, a single-center retrospective study described the clinical characteristics of 42 patients with NTBE [8]. The most underlying disease was malignancy (17 cases, 40.5%), followed by antiphospholipid antibody syndrome (15 cases, 35.7%) and systemic lupus erythematosus (14 cases, 33.3%). Most of these malignancies included lung cancer, breast cancer, and pancreatic cancer, while none were associated with acute leukemia. Almost all patients were treated with anticoagulation, as in our case. However, the mortality rate was as high as 35%, and among them, patients with malignancies had a significantly worse prognosis than those with autoimmune diseases. Patients with malignancies often had a poor prognosis because their general conditions were already poor [1,8].

The case reports of NTBE associated with acute leukemia are extremely rare. Most of these patients were diagnosed by pathological autopsy [3]. A previous study reported that in the past, one patient with newly diagnosed APL developed NTBE [9]. Due to clinically suspected IE, antibiotics were first administered. However, the patient did not respond to antibiotics and underwent surgical removal of vegetation from the aortic valve. The pathological finding indicated fibrin formation with very few inflammatory cells; hence, the diagnosis of NTBE was confirmed postoperatively. Moreover, one patient with acute myeloid leukemia was reported to have expired during induction therapy and was diagnosed with NBTE, cerebral infarctions, and DIC based on the pathological autopsy [10]. These findings of vegetation and thrombi also showed fibrin formation. Unlike them, from autopsy findings obtained from our case, the large number of leukemic cells was observed in the vegetation and thrombi, suggesting that her disease status of APL was more severe at the first visit. Furthermore, since the patient had already suffered a stroke and her white blood cell and platelet counts were almost within normal range, it would have been extremely difficult to definitively diagnose both APL and NTBE during her lifetime.

In some cases where surgery was performed for NTBE, the prognosis was good [8]. Moreover, in the previously reported cases of NTBE associated with APL, surgical treatment was followed by combination therapy with all-trans retinoic acid (ATRA) and arsenic trioxide, and the general condition improved [9]. Anticoagulation therapy is relatively contraindicated in cases of DIC complicated by APL, therefore, immediate treatment with ATRA is recommended [11,12]. In fact, there is a report of a case of NBTE caused by APL that was treated with arsenic trioxide and ATRA, without the administration of anticoagulants [13].

A report showed that automated blood cell counting systems could lead to the misdiagnosis of APL [14]. In addition, the fact that the patient was one of our colleagues and the prime time for treatment of stroke was approaching led to “premature closure” of the diagnosis [15].

If we were aware of these precedents, a correct diagnosis would have been earlier established, and the treatment would have been changed to clot retrieval and APL treatment, without the administration of anticoagulants. Our present case could have been saved if there had been adequate time to establish a correct diagnosis, the vegetation was surgically removed, and induction therapy with ATRA was performed.

This case prompted a discussion about equipment optimization, awareness of the limitations of automated cell counters, specialist development, and equitable medical care regardless of working hours in a rural hospital like ours. After her death, the M&M conference was held to share the case experience. Consequently, the doctors lectured all medical staff to facilitate a deeper understanding of APL, and improvements have been made in the proper assignment of laboratory technicians and the panic value reporting system.

## 5. Conclusions

In this study, we experienced a fatal case of cerebral infarction caused by NBTE with sudden onset of neurological symptoms. We herein present the first case in the world where the diagnosis of APL-associated NTBE was established by autopsy. NBTE can be challenging situation for physicians because of the wide range of differential diagnoses. APL should be considered as a differential diagnosis of NBTE with coagulopathy and should not be readily treated with anticoagulants, even if it causes embolization.

## Figures and Tables

**Figure 1 medicina-57-01264-f001:**
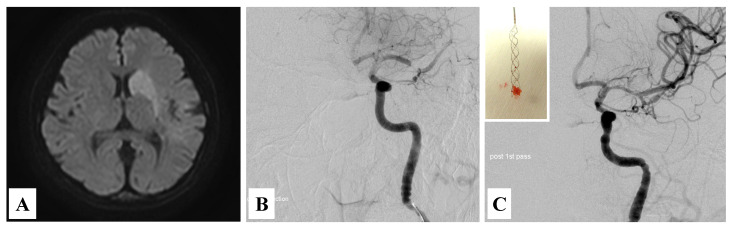
Images of cerebral occlusion and endovascular thrombectomy in this patient with neurological symptoms when visiting to our hospital. The brain MRI showed a high-signal-intensity change in her left brain (**A**). Angiography indicated the occlusion of the left M1 portion before the endovascular thrombectomy (**B**). After the thrombectomy, almost complete recanalization was confirmed (**C**). Inset in (**C**) showed the arterial occlusive thrombus.

**Figure 2 medicina-57-01264-f002:**
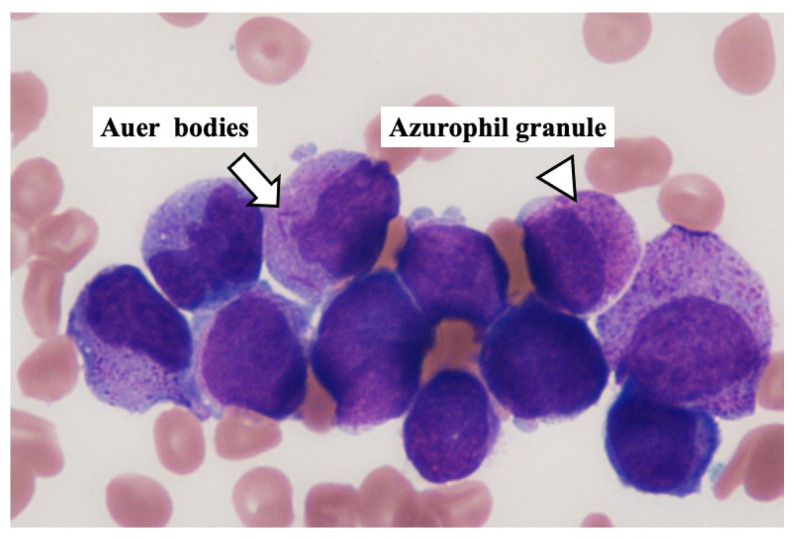
Leukemia cells in the peripheral blood obtained from this hospitalized patient. The peripheral blood smear shows a possible presence of myeloid leukemia cells with Auer rods (arrow) and azurophilic granules (arrowhead).

**Figure 3 medicina-57-01264-f003:**
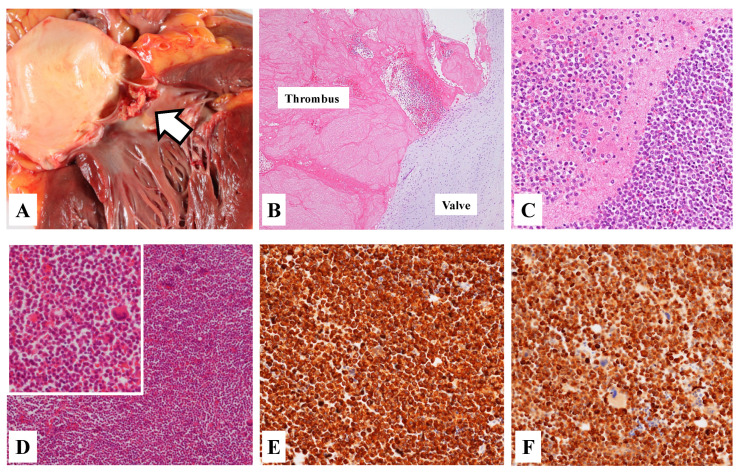
Macroscopic and microscopic appearances in an autopsy of our patient. Macroscopic finding of the heart showed the vegetation (**A**, arrow). Histological finding revealed the evidence of the thrombus attached to the arterial valve (**B**, H&E staining). Moreover, the close-up finding showed the aggregation of abnormal leukocyte cells in the thrombus (**C**, H&E staining). Histological finding of the femoral bone marrow showed massive abnormal leukocyte cells (**D**, H&E staining). From immunohistochemistry method, most abnormal cells were positive to myeloperoxidase (MPO) (**E**) and MIB-1 (**F**).

**Table 1 medicina-57-01264-t001:** Laboratory data of the patient on admission to the hospital.

Complete Blood Count	Results	(Normal Range)
White blood cells	6.8 × 10^9^/L	(3.59.9)
Hemoglobin	11.0 g/dL	(11.0–14.8)
Mean corpuscular volume	96.4/fL	(83–99)
Platelets	100 × 10^9^/L	(120–400)
Blood Coagulation Tests		
PT%	57%	(80–120)
APTT	33.2 s	(24.0–39.0)
Biochemistry		
Total protein	6.7 g/dL	(6.5–8.2)
Albumin	3.8 g/dL	(3.5–5.0)
Total bilirubin	0.34 mg/dL	(0.2–1.2)
Glutamic oxaloacetic transaminase	21 IU/L	(5–40)
Glutamic pyruvic transaminase	12 IU/L	(3–35)
Lactate dehydrogenase	401 IU/L	(124–222)
Blood urea nitrogen	15.8 mg/dL	(8.0–23.0)
Creatinine	0.75 mg/dL	(0.62–1.10)
Sodium	140 mEq/L	(139–146)
Potassium	3.7 mEq/L	(3.7–4.8)
Chloride	106 mEq/L	(101–109)
Alkaline phosphatase	203 IU/L	(38–113)
C-reactive protein	3.09 mg/dL	(0.0–0.20)

Abbreviations: APTT, activated partial thromboplastin time; PT%, prothrombin time.

## Data Availability

All data are included in the main text.

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
