# Peer review of "Nonbacterial Thrombotic Endocarditis Associated with Acute Promyelocytic Leukemia: An Autopsy Case Report"

_medicina, 2021, doi:10.3390/medicina57111264_

Round 1

Reviewer 1 Report

Very interesting case report describing non-bacterial thrombotic endocarditis as a presenting feature of acute promyelocytic leukemia. 1. Although I agree that the blood cell counts were almost normal in this case, most currently available automated cell counters do flag abnormal cells while processing complete blood counts on peripheral blood. Generally this triggers a peripheral smear examination leading to further tests for confirming the diagnosis of acute promyelocytic leukemia The present case highlights the importance of having a facility for a reliable peripheral smear examination even in during non-regular times in the hospital. 2. Secondly, there is a previous report of treating non-bacterial thrombotic endocarditis in acute promyelocytic leukemia non-surgically using prednisolone, arsenic trioxide and all-trans retinoic acid. (L Kuffer, M I Koerber, F Nettersheim, T Tichelbaecker, C Hohmann, A K Schaetzle, K Kabbasch, J G Borrega, B Boell, F Kohle, C Warnke, S Baldus, H Ten Freyhaus, P630 Recovery of non-bacterial thrombotic endocarditis and severe aortic regurgitation in a young patient with acute promyelocytic leukemia, European Heart Journal - Cardiovascular Imaging, Volume 21, Issue Supplement_1, January 2020, jez319.314, https://doi.org/10.1093/ehjci/jez319.314).

Reviewer 2 Report

This is an interesting case report depicting a diagnostic error.

The discussions must be improved. The presence of bicytopenia should have been investigated with at least the following: leukocyte formula (could have mentioned the presence of promyelocytes and of the leukemic hiatus characteristic of AML and could have oriented the diagnosis towards APL), a peripheral blood smear (schistocytes would have been present as characteristic feature of microangiopathic hemolytic anemia in the context of APL-related disseminated intravascular coagulation), fibrinogen levels, fibrin monomers, D-dimers as markers of secondary fibrinolysis. In the presence of bicytopenia and altered Quick time a hematology consultation would have been necessary as the suspicion of APL could have saved the life of the patient. The intracerebral hemorrhage was probably caused by the use of anticoagulants in the context of DIC. These aspects should all be discussed and a clear strategy for such cases should be presented so that this diagnostic error will not be repeated by others as APL is a hematological emergency.

Add the reference values of your lab in a separate column in Table 1. Revise the Table to match the style of the journal.

Revise the references to match the style of the journal - American Chemical Society.

Round 2

Reviewer 2 Report

The authors have addressed my comments and the paper can be accepted for publication.